# Phosphorylation deficient inducible cAMP early repressor (ICER) modulates tumorigenesis and survival in a transgenic zebrafish (*Danio rerio*) model of melanoma

Justin Wheelan, Melissa Spigelman, Angelo Cirinelli, James Reilly and Carlos A. Molina*

## ABSTRACT

Melanoma, the most lethal form of skin cancer, is commonly associated with mutations in the BRAF gene, particularly BRAF$^{V600E}$, which drives tumor proliferation via the ERK1/2 signaling cascade. While BRAF inhibitors initially demonstrate efficacy, therapeutic resistance remains a significant challenge. Emerging evidence implicates the cAMP signaling pathway, particularly the cAMP response element-binding protein (CREB) and its repressor, inducible cAMP early repressor (ICER), in melanoma progression and drug resistance. ICER, a transcriptional repressor regulated via Ras/MAPK-mediated phosphorylation and ubiquitination, is degraded in melanoma, undermining its tumor-suppressive role. In a *braf*$^{V600E}$; *p53* (loss of function) transgenic zebrafish (*Danio rerio*) model, we investigated the role of a ubiquitin-resistant ICER mutant (S35-41A-ICER) in tumor progression. Transgenic fish expressing S35-41A-ICER exhibited extended survival and reduced tumor invasiveness compared to wild-type ICER. RNA sequencing revealed dysregulation of CREB/CREM targets and compensatory pathways, including Rap1 and PI3K/AKT signaling, as well as candidate gene targets of ICER regulation, including the protein kinase A catalytic subunit *prkacaa*. Our findings suggest that a ubiquitin resistant ICER mitigates melanoma progression and represses oncogenic pathways in a *braf*$^{V600E}$ melanoma context.

KEY WORDS: Melanoma, Cancer, ICER, CREM, CREB, PKA

## INTRODUCTION

Melanoma, originating from pigment-producing melanocytes, remains the most lethal form of skin cancer despite its relative rarity compared to other cutaneous malignancies (Cancer.gov). In 2024, an estimated 100,640 new cases of invasive melanoma were diagnosed and 8290 associated deaths were reported in the USA (Cancer.org, 2024). Approximately 70–80% of melanocytic nevi and ∼50% of malignant melanomas harbor mutations in the *BRAF* gene, with BRAF$^{V600E}$ being the most prevalent alteration. This mutation leads to constitutive activation of the extracellular signal-regulated kinase (ERK1/2) cascade, driving tumor proliferation and survival (Davies et al., 2002; Wan et al., 2004). Although pharmacological inhibition of BRAF$^{V600E}$, using agents such as vemurafenib, has shown efficacy

in reducing tumor burden, these responses are often transient due to the development of therapeutic resistance (Bollag et al., 2012; Chan et al., 2017). Immune checkpoint inhibitors, including anti-PD1 (nivolumab, pembrolizumab) and anti-CTLA-4 (ipilimumab), have revolutionized melanoma treatment, especially in late-stage metastatic disease, increasing long-term survival to 50% from less than 10% historically (Carlino et al., 2021). Despite these advances, treatment resistance continues to pose significant challenges, underscoring the urgent need for alternative therapies (Ascierto et al., 2024; Winder and Virós, 2018).

Emerging evidence highlights the importance of the cyclic AMP (cAMP) signaling pathway in melanoma progression and resistance, despite infrequent genetic mutations within this pathway (Johannessen et al., 2013; Gough, 2013). Transcriptional regulation downstream of cAMP is mediated following phosphorylation of cAMP response element-binding protein (CREB) and other members of cAMP responsive proteins such as, cAMP responsive element modulator (CREM), which includes activators and repressors that bind cAMP-response elements (CREs) promoter motifs via basic leucine zipper domains. One way the activation of these kinase inducible proteins occurs, is through cAMP induced activation of protein kinase A (PKA). Secondary messenger activity via the cAMP-PKA signaling pathway has been implicated in numerous processes relevant to tumor biology, such as glycogen metabolism, ion channel regulation, cell differentiation and proliferation and gene induction (Rohlff and Glazer, 1995). Resistance to BRAF and MEK inhibitors in BRAF$^{V600E}$ melanoma has been linked to compensatory activation of cAMP-dependent signaling networks, involving PKA, and CREB (Johannessen et al., 2013). Biopsies of BRAF$^{V600E}$ melanoma suggest that phosphorylated (and therefore active) CREB is suppressed by RAF-MEK inhibition, but restored in relapsing tumors (Johannessen et al., 2013). We hypothesize that inducible cAMP early repressor (ICER), an antagonist of CREB and CREM in this context, may play a critical role in melanoma resistance and tumor progression.

ICER arises from an intronic promoter of the *CREM* gene and exists as four isoforms: ICER-I, ICER-Iγ, ICER-II, and ICER-IIγ, all of which retain a DNA-binding domain but lack the kinase-inducible and N-terminal domains present in other CREB family members (Stehle et al., 1993; Molina et al., 1993). Despite its compact structure, ICER functions as a potent transcriptional repressor, acting as a homodimer or heterodimer with other CREB family members to autoregulate its own expression by binding the prototypical palindromic CRE sequences (5′ TGACGTCA 3′) within its promoter (Molina et al., 1993). Evolutionarily conserved, ICER retains ∼85% sequence identity across species, underscoring its critical role as a nuclear effector of cAMP signaling.

ICER's regulatory significance extends to diverse physiological processes, including circadian rhythm (Takahashi, 1994), cardiac myocyte regulation (Tomita et al., 2003), cell cycle regulation

Department of Biology, Montclair State University, Montclair, NJ 07043, USA.

*Author for correspondence (molinac@montclair.edu)

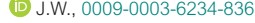 J.W., 0009-0003-6234-8360

(Yehia et al., 2001), and as a tumor suppressor (Pigazzi et al., 2008; Mémin et al., 2002). However, its degradation and subcellular localization are tightly controlled via post-translational modifications. More recently, CREM, has also been identified as a transcriptional checkpoint in natural killer (NK) cells, where its deletion enhances CAR-NK cell anti-tumor efficacy by derepressing key effector genes such as *PRF1* and *GZMB*, implicating this pathway in broader immunoregulatory functions (Rafei et al., 2025).

Our prior work demonstrated that ICER proteasomal degradation is influenced by Ras/MAPK-mediated phosphorylation and subsequent ubiquitination. Using Tyr/Tet-Ras *INK4a*−/− transgenic mice and melanoma-derived R545 cells, we showed that ICER degradation is dependent on oncogenic H-RasV12G expression (Mémin et al., 2011). Pharmacological inhibition of Ras or the proteasome abolished ICER degradation, highlighting the role of Ras-driven proteasomal regulation (Healey et al., 2013). Further studies revealed that ICER is contextually regulated by phosphorylation at serine's 35 and 41. Phosphorylation at serine 35 leads to ICER monoubiquitination and cytosolic relocalization (Healey et al., 2013), while phosphorylation at serine 41 triggers polyubiquitination and proteasomal degradation (Mémin et al., 2011). The latter process is mediated by the E3 ubiquitin ligase UBR4, operating via the N-end rule pathway and requiring recognition of ICER's N-terminus (Cirinelli et al., 2022). Our prior studies revealed that a modified N-terminal ICER tagged with human influenza hemagglutinin (NHA-ICER) induced apoptosis five times more effectively than C-terminal HA-tagged ICER, despite no significant differences in DNA-binding affinity between the two variants (Cirinelli et al., 2022).

In this study, we aimed to investigate the impact of ICER phosphorylation and subsequent ubiquitination on melanoma progression using a *braf*$^{V600E}$ +/+, *p53* (loss-of-function), and *mitf* (loss-of-function) transgenic zebrafish (*Danio rerio*) model. In this model we employ the miniCoopR system, where *mitf* expression is restored only upon successful transgene integration. Using this model, we generated three distinct transgenic cohorts: (1) ubiquitin-resistant ICER with serine-to-alanine substitutions at positions 35 and 41, and an N-terminal hemagglutinin (HA) tag (S35-41A-ICER); (2) wild-type ICER with an N-terminal HA tag (wtICER); and (3) an enhanced GFP (EGFP) control. We evaluated overall survival of these cohorts over 104 weeks. Tumor histology was assessed via Hematoxylin and Eosin (H&E) staining. Additionally, tumor-derived cell lines were established to perform RNA sequencing and investigate ICER's role in transcriptional regulation. Our findings demonstrate that fish expressing the ubiquitin-resistant S35-41A-ICER mutant exhibit extended survival compared to wtICER and EGFP controls. Histological analysis revealed that tumors expressing wtICER were the most aggressive and invasive, while tumors in the S35-41A-ICER cohort appear to have markedly reduced invasiveness, though it is not clear if the tumors are melanomas or a soft tissue sarcoma. Cell lines derived from wtICER tumors showed accelerated proliferation compared to EGFP controls, consistent with ICER's inability to exert its repressive function due to cytosolic relocation. Interestingly, we were unable to establish stable cell lines from S35-41A-ICER tumors, likely due to its suppression of cell proliferation, reinforcing its role as a tumor suppressor. RNA sequencing revealed a set of differentially expressed genes between wtICER and EGFP cell lines, many of which overlap with known CREB/CREM target. Gene Ontology (GO) of differentially expressed genes suggests compensatory rewiring of cellular

networks in response to ICER dysregulation leading to an increase in cell proliferation, focal adhesion, and pro-metastatic signaling pathways, such as Rap1 and PI3K/AKT. Finally, we also propose a novel mechanism, in which ICER may directly reduce cAMP activation of PKA and subsequent CREB phosphorylation via competitive CRE binding on the *prkacaa* promoter, encoding the Protein Kinase A catalytic α-subunit. In this article, we highlight ICER's dual role as a repressor of oncogenic pathways and a target of proteasomal degradation and show that the ubiquitin-resistant S35-41A-ICER mutant not only mitigates tumor progression but also prolongs survival in a *braf*$^{V600E}$-driven melanoma context.

## RESULTS
### Expression of ICER protein is abnormal in zebrafish melanomas
To validate and extend previous findings from Yehia et al. (2001) and Healey et al. (2013), which demonstrated post-translational regulation of ICER in human melanoma cells and a mouse model of melanomagenesis, we sought to assess ICER protein expression in a zebrafish model. Endogenous ICER protein levels were examined in tissue samples collected from either normal skin or melanomas of seven individual zebrafish. As shown in Fig. 1A, ICER protein expression was markedly reduced in tissues containing melanomas (lanes 3–7). In contrast, robust ICER expression was observed in normal skin tissues from both wild-type AB fish and non-tumor transgenic fish (lanes 1 and 2).

To control for the presence of other skin cell types that may confound interpretation of total tissue lysates, we also performed western blot analysis on FACS-sorted *mitfa*$^+$ skin cells, which include a mixture of melanocytes and xanthophores but exclude other epidermal and stromal populations. As shown in Fig. 1B, ICER protein was not detectable in the *mitfa*$^+$ cell population isolated from either 6- or 14-month-old fish. While we did not assess ICER mRNA levels in these samples and so transcriptional downregulation cannot be ruled out, the absence of detectable protein likely reflects post-transcriptional or post-translational mechanisms. This interpretation is consistent with prior observations showing reduced ICER protein despite unchanged mRNA levels in both melanoma cell lines and tumor tissues (Healey et al., 2013).

Together, these results support the hypothesis that ICER is subject to aberrant degradation or translational repression in tumorigenic contexts. Given that melanomas in this zebrafish model arise through melanocyte-specific expression of braf$^{V600E}$, a mutation that constitutively activates the MAPK pathway, these findings further implicate ICER in the suppression of melanoma initiation and maintenance of the transformed phenotype.

### ICER expression impacts tumorigenesis and survival in *braf* V600E, *p53* loss-of-function, and *mitf* loss-of-function transgenic zebrafish
We then tested whether autochthonous ICER expression in melanocytes would impact the survival of zebrafish with a constitutively active *braf*$^{V600E}$ mutation, in the context of a loss of function of *p53* gene, and with *mitf* rescued via the Tol2 integration of the miniCoopR plasmid, as described in Ceol et al. (2011) and Iyengar et al. (2012). Pairwise *t*-test suggests a significant (P<0.05) decrease in survival in the fish, expressing wtICER (n=48) with an N-terminal HA-Tag, and demonstrable lower median and overall survival compared to EGFP control (n=37) (Fig. 2). Simultaneously, we evaluated the impact of the phosphorylation deficient ICER mutant (S35-41A-ICER) (n=40), also containing an

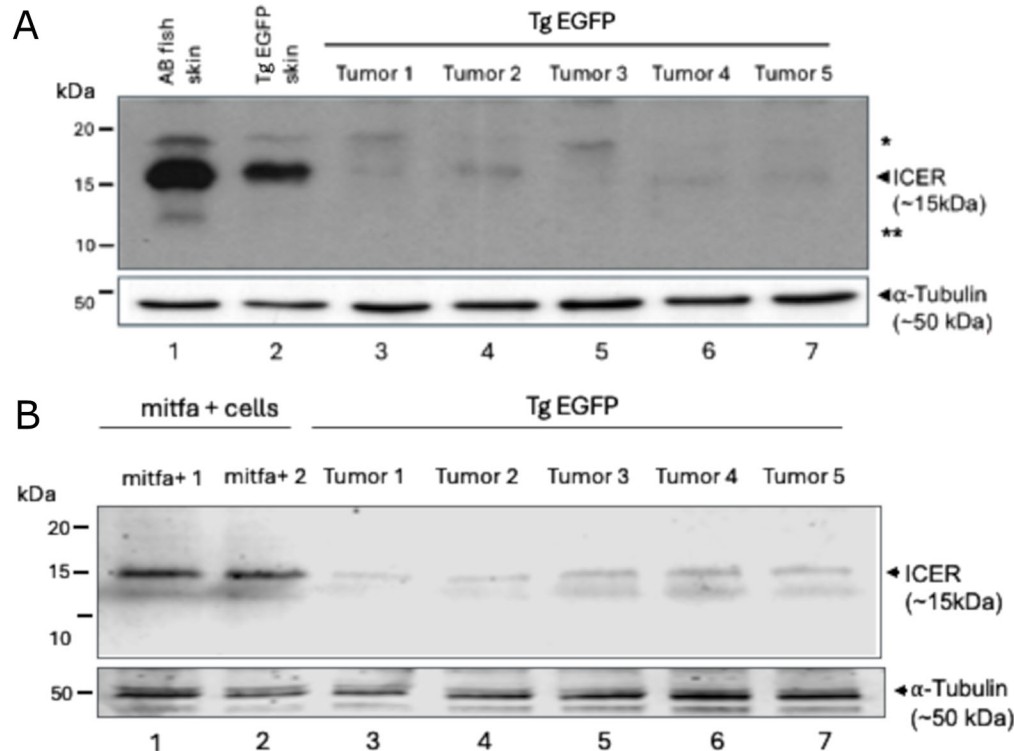

**Fig. 1. ICER protein expression in zebrafish tissues.** (A) Western blot analysis of protein expression in melanomas from the indicated zebrafish strains. Tissue samples were collected from skin and tumor tissues as shown. Western blotting for endogenous ICER expression was performed from whole tissue extracts using anti-ICER polyclonal antibodies (Cirinelli et al., 2022). The same membrane was stripped and re-probed with anti-α-Tubulin antibodies as a loading control (bottom panel). The left margin indicates the relative mobility of low range molecular weight markers from Bio-Rad (in kDa). The right margin indicates the relative mobility of ICER and α-Tubulin. Asterisks denote post-translationally modified forms of ICER (Mémin et al., 2011; Cirinelli et al., 2022). (B) Western blot analysis of FACS-sorted *mitfa*$^+$ skin cells isolated from *mitfa:mCherry*$^+$; *crestin:EGFP*$^+$; *tyrosinase*$^{-/-}$ zebrafish at 6 and 14 months post fertilization. Skin tissue was manually dissected, enzymatically dissociated, and filtered to generate a single-cell suspension. Live *mitfa*$^+$ cells (DRAQ7$^-$, mCherry$^+$) were isolated via FACS. These *mitfa*$^+$ skin cells include a mixed population primarily consisting of melanocytes and xanthophores. A total of 98,000 sorted cells pooled from 5–7 zebrafish were lysed in RIPA buffer supplemented with protease inhibitors. Lysates were clarified and subjected to western blotting as described in (A). ICER protein was not reduced in the tumor derived cells. While the absence of detectable protein may reflect post-transcriptional instability or proteasomal degradation of ICER, the possibility of transcriptional downregulation cannot be excluded, as mRNA levels were not assessed in this experiment.

N-terminal HA-tag. Conversely, this isoform rescued fish survival compared to the wtICER ($P<0.05$). However, while the S35-41A-ICER has a notably longer lifespan than either wtICER or EGFP control cohorts, the difference between S35-41A-ICER and EGFP control in this study was not statistically significant via pair-wise *t*-test ($P>0.05$).

Histological evaluation of tumor progression and invasiveness was performed using H&E staining to visualize cellular and tissue architecture. To assess tumor invasiveness, histological sections were collected 2 weeks after the first lesion was observed, and reviewed for evidence of muscle invasion, defined as tumor cell infiltration beyond the dermal boundary into adjacent skeletal muscle fibers, as judged by disruption of the dermal-muscular interface and cellular infiltration patterns. HA-wtICER group: five out of six fish showed clear muscle invasion. EGFP control group: none of four evaluable tumors showed invasion. HA-S35&41A ICER mutant group: tumors from more than six independent fish were evaluated across replicate experiments, and none showed evidence of muscle invasion. While only one high-resolution representative image is included in the main Fig. 3, this image reflects the best histological section available. All findings were confirmed across multiple independent samples and timepoints. The pathological analysis of the tumors shows that melanomas from

EGFP-expressing fish show exophytic growth without invading the underlying musculature as shown before (Ceol et al., 2011). In contrast, the melanomas from wtICER-expressing fish invaded from the skin, through the collagen-rich stratum compactum of the dermis, into the underlying musculature. Fish expressing S35-41A-ICER do not appear to show malignancies, showing signs of benign hyperplasia, further supporting the hypothesis that the ubiquitin-resistant ICER variant mitigates tumor progression and invasion. However, more work is needed to fully characterize the growth observed in the S35-41A-ICER fish. Additionally, we have not been able to isolate the S35-41A ICER protein, so we cannot fully rule out that the protein is not stably expressed.

## Generation and characterization of zebrafish tumor-derived cell lines

To address the challenge of tumor heterogeneity in bulk RNA-seq experiments and to enable a more controlled analysis of transcriptional changes, we generated stable cell lines from tumors excised from transgenic zebrafish cohorts. Tumors were surgically extracted, enzymatically dissociated into single-cell suspensions, and cultured under conditions optimized for zebrafish cells (see Materials and Methods). These cell lines were derived from MiniCoopR-EGFP (control) and MiniCoopR-HA-wt-

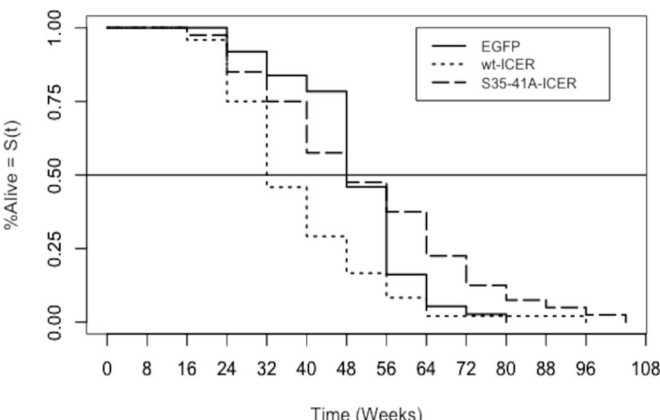

**Fig. 2. Survival Analysis.** Kaplan–Meier survival curve comparing percent survival among three transgenic zebrafish cohorts: EGFP (*n*=37), wtICER (*n*=48), and S35-41A-ICER (*n*=40) over a 104-week observation period. Survival curves were generated using the Kaplan–Meier estimator, with survival differences assessed using the log-rank test (*P*<0.05). Fish expressing wtICER displayed significantly reduced survival compared to the EGFP control via pair-wise *t*-test, correlating with more aggressive tumor histology observed in Fig. 3. In contrast, fish expressing the S35-41A-ICER mutant showed improved survival compared to wtICER (*P*<0.05), consistent with reduced tumorigenesis.

ICER transgenic zebrafish, referred to as 446A and 447A, respectively. Brightfield and fluorescence microscopy confirmed the successful establishment of the EGFP cell line, with robust EGFP expression observed in 446A cells (Fig. 4A). PCR analysis verified the integration of the HA-ICER transgene into the genome of the 447A cell line (Fig. 4B), and western blotting further confirmed the expression of HA-tagged ICER protein in 447A cells (Fig. 4C). Notably, the presence of the N-terminal HA-tag likely

prevents recognition by the ubiquitin-proteasome system, thereby stabilizing ICER protein and enabling its detection in this context. Immunocytochemical staining of 447A cells revealed nuclear and cytosolic localization of the HA-ICER protein, consistent with its function as a transcription factor and in conjunction with known delocalization of mono-ubiquitinated ICER from the nucleus (Healey et al., 2013) (Fig. 4D).

## Differential gene expression and pathway enrichment analysis reveals tumorigenic profile of wtICER

Using the previously validated cell lines derived from the zebrafish tumors, we sought to examine the transcriptomic profiles of the tumors. Three independent cultures per genotype were grown until about $1×10^6$ cells per dish and total RNA was isolated to understand not only differentially expressed coding genes (DEGs), but also other non-coding RNAs that may be present in the sample. We found 2284 significantly (*P*<0.05) differentially expressed genes between the EGFP and wtICER fish tumor cells. DEGs were then analyzed using ShinyGO for GO, which suggests pathways associated with more aggressive tumors in the wtICER were differentially expressed, in agreement with previously observed survival and histological studies. Of note, we observed differential expression of genes involved in glycosaminoglycan keratan sulfate biosynthesis (Fig. 5A), which has been implicated in melanoma and other cancers to contribute to focal adhesion and enhanced invasiveness (Tachibana et al., 2022; Leiphrakpam et al., 2019). Specifically, related to this pathway, we observed down-regulation of genes such as *b4galt3*, *st3gal1*, and a notable 5-fold increase in *b3gnt7* expression, a combination that is consistent with current metastasis literature in many cancer types (Chen et al., 2023; Xie et al., 2024). Other pathways shown in Figs 5B and 6A, contain overlapping genes, that have considerable literature about their interactions and relation to many types of cancer as well as melanoma, including but not limited to *akt1* (Cho et al., 2015), *cdkn1a* (Jalili et al., 2012), *col4a4* (Shin et al., 2022), *fgf5*

miniCoopR-EGFP          miniCoopR-wt-HAICER          miniCoopR-HAS35/41A-ICER

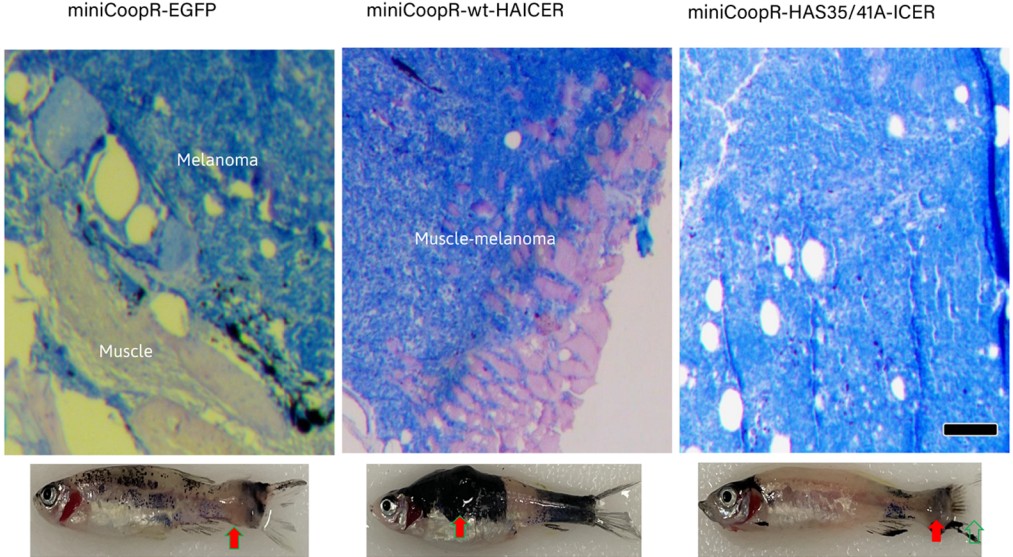

**Fig. 3. Tumor histology and gross phenotypic analysis of transgenic zebrafish cohorts expressing EGFP, wtICER, and S35-41A-ICER.** Representative images of zebrafish and corresponding H&E-stained tumor sections for each cohort. From left: EGFP-expressing zebrafish exhibit melanomas localized to the skin and minimal infiltration into proximal muscle tissue. Middle: wtICER-expressing zebrafish display extensive tumor growth with pronounced infiltration into the underlying muscle, indicative of an aggressive melanoma phenotype. Right: S35-41A-ICER expressing zebrafish shows some cellularity, though presents distinctly from the other samples. Gross phenotypic analysis mirrors these histological findings, with fish expressing EGFP showing moderate tumor size, wtICER fish exhibiting large, and widespread tumors, and S35-41A-ICER fish displaying smaller lesions. While the exact type of tumor was not able to be confirmed, it is believed to be melanoma. Scale bar: 70 µm.

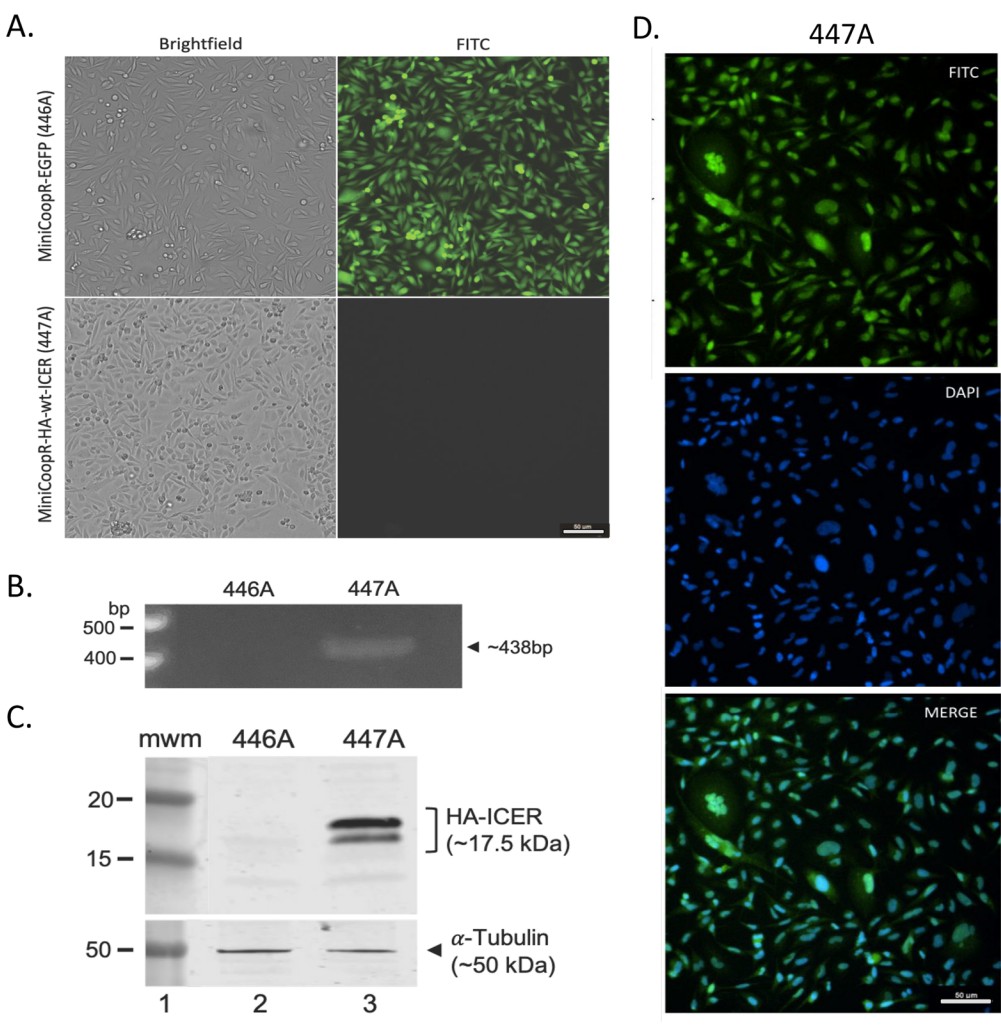

**Fig. 4. Characterization of established cell lines from fish melanomas.** (A) Images of brightfield and fluorescence microscopy of MiniCoopR-EGFP (446A) and MiniCoopR-HA-wt-ICER (447A) cell lines in culture. (B) PCR analysis of 446A and 447A cell lines to test for HA-ICER transgene integration into the fish genome. (C) Anti-HA Western blot analysis of 446A and 447A cell lines to test for transgenic HA-ICER protein expression. 446A was used as a control. (D) Immunocytochemical determination of the subcellular localization of the transgenic HA-ICER protein in 447A cell line. 446A was used as a control, showing no signal (not shown). Commercially available anti-HA antibodies were used in the experiment shown in C and D, and anti-αTubulin antibodies in C as loading control. Scale bar: 50 μm.

(Ghassemi et al., 2017), *fgfr2* (Gartside et al., 2009), *met* (Zhou et al., 2019), *notch3* (Howard et al., 2013), *pdgfra* (Sabbatino et al., 2014), *pdgfrb* (Kadrmas et al., 2020), as well as upregulation of the proto-*oncogene junb*, which has also been shown to increase tumor growth and metastasis in murine models, and has been associated with poor outcomes in clinical breast and ovarian cancer patients (Pérez-Benavente et al., 2022).

We also note an 8-fold increase in expression of the growth differentiation factor 6 *gdf6a*, which has been shown to be transcriptionally upregulated in melanoma and is involved in BMP signaling, promoting a neural crest signature and impacting differentiation (Venkatesan et al., 2018).

### Mechanistic investigation of ICER function suggests a link to auto-regulatory interaction with critical genes involved in melanogenesis

We then sought to elucidate the mechanisms contributing to the more aggressive melanoma phenotype observed in the wtICER transgenic zebrafish compared to the EGFP control and phosphorylation-deficient mutant cohorts. To start, we sorted our DEGs for significantly upregulated genes ($P<0.05$) in wtICER compared to EGFP, consistent with the hypothesis that gene upregulation in this context could result from dysregulation of ICER as a transcriptional repressor. The promoter regions of these genes were analyzed for full CREs (5′ TGACGTCA 3′) and half-CREs

(5′ TGACG 3′/5′ CGTCA 3′) using find individual motif occurrences (FIMO) via MEME-suite (version 5.5.7) with a threshold of $P<0.001$ (Grant et al., 2011). This yielded a list of candidate genes that ICER may regulate through DNA binding at CRE sites.

We also created bulk RNA-seq libraries from solid neoplastic tissue for each respective cohort: EGFP ($n=6$), wtICER ($n=6$), and the phosphorylation deficient S35-41A-ICER ($n=6$). The expression patterns were expectedly heterogeneous, but helpful to directionally identify and filter for genes that were expressed at lower levels in the phosphorylation-deficient ICER tumors compared to wtICER. Drawing on prior studies in murine and human systems that established conserved ICER mechanisms and function, we reasoned that pertinent pathways and gene regulation would also exhibit conservation in this zebrafish model. To explore translational relevance, we identified the orthologous human gene IDs (hg38) for zebrafish genes using Ensembl biomart (Harrison et al., 2024) in our list and repeated the promoter analysis as described above. This cross-species analysis produced 58 genes with CRE motifs present in both zebrafish and human promoter sequences. These genes were upregulated in wtICER fish but not in the EGFP or phosphorylation-deficient ICER fish, suggesting a potential role for ICER dysregulation in their transcriptional activation.

To further support the functional relevance of these genes, we followed these analyses by analyzing publicly available chromatin immunoprecipitation and sequencing (ChIP-seq) data from the

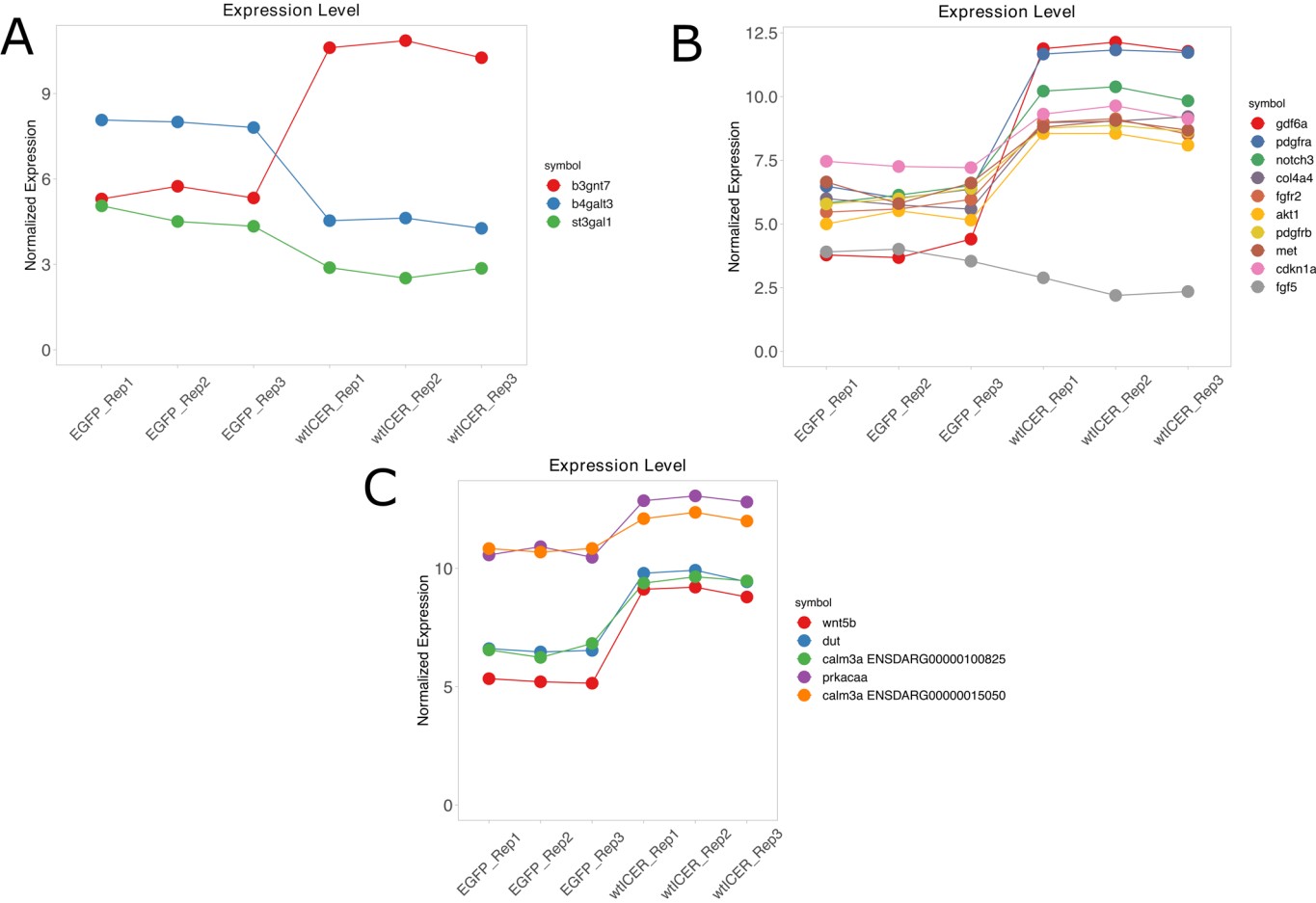

**Fig. 5. Gene expression comparison.** Expression signature of wtICER (*n*=3) and EGFP (*n*=3) of relevant genes involved in (A) Glycosaminoglycan biosynthesis of keratan sulfate. (B) Genes related to numerous tumor biology pathways, found to be differentially expression in this experiment. (C) Genes found to be differentially expressed, based on DAVID query of possible ICER regulatory genes via GO, found to be involved in melanogenesis. Differential gene expression analysis was conducted via iDEP (version 2.0.1) with FDR ≤0.05, and log2 (fold-change) ≥2.

ENCODE project. Specifically, we examined datasets targeting CREM in two distinct human cell lines: K-562, a chronic myeloid leukemia (CML) line derived from the pleural effusion of a 53-year-old woman with CML in blast crisis, and HEP-G2, a hepatocellular carcinoma line established from the tumor of a 15-year-old boy. Of the 58 candidate genes, 20 were found to overlap with CREM ChIP-seq peaks in the K-562 cell line, and 31 overlapped in the HEP-G2 cell line (Fig. 6B). We find 16 genes that overlap in both datasets (*crema, zcchc24, prkacaa, iqsec1b, rgs12b, sik2b, gli2a, gab1, ttyh3b, gyg1b, zbtb16b, actn4, junba, plxdc2, plpp3,* and *hmga2*). Concordant ChIP-SEQ results coupled with our RNA-seq analysis provides evidence of ICER regulation, and antagonism of CREM/CREB, however, absent melanoma specific ChIP-seq data targeting CREM, it is unclear how cell type specific chromatin accessibility may impact ICER/CREM binding.

GO analysis of the 16 candidate genes using DAVID (Sherman et al., 2022) revealed four genes *PRKACA, DCT, CALM3,* and *WNT5B* with key roles in the regulation of melanogenesis (Fig. 5C). These genes may be directly regulated by ICER via upstream binding to CRE motifs. Using UCSC genome browser, we manually confirmed that the promoter region for each gene contained CRE motifs (including half-CRE's) based on the FIMO output, for both zebrafish GRCz11 and human hg38 genomes.

## DISCUSSION

Although advancements in treatment such as immunotherapies and targeted therapies have improved patient outcomes in melanoma, there remains an urgent need for additional therapeutic strategies to address resistance and improve long-term survival. BRAF, and other MAPK pathway inhibitors, while initially highly effective at reducing tumor burden, are subject to drug resistance. One compensatory mechanism implicated in resistance to MAPK pathway inhibition, such as through BRAF and MEK inhibitors, is the activation of the cAMP signaling pathway. Intracellular cAMP levels can be elevated via G-protein-coupled receptor (GPCR) activation of adenylyl cyclase or through the inhibition of phosphodiesterase, which degrades cAMP. Increased cAMP activates PKA, which phosphorylates various downstream targets, including members of the CREB protein family. ICER, a potent transcriptional repressor and antagonist to CREB/CREM, plays a pivotal role in regulating tumorigenesis and has potential to mitigate resistance to BRAFi therapies. However, endogenous ICER's therapeutic utility is inherently limited by its susceptibility to post-translational modification. Phosphorylation at serine 41, mediated by MAPK, triggers polyubiquitination and proteasomal degradation, a process regulated by the N-end rule pathway and requiring recognition by the ubiquitin ligase UBR4, however

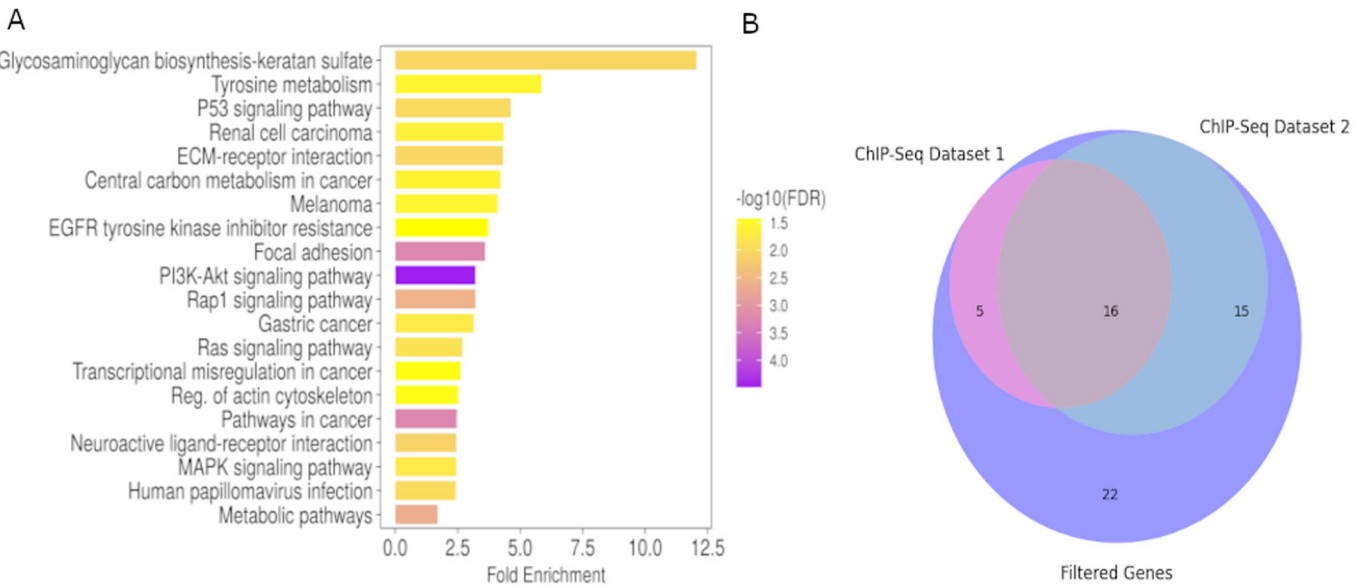

**Fig. 6. Pathway enrichment.** (A) Bar chart of ShinyGO output of all DEG from the zebrafish tumor cell lines for EGFP (*n*=3) and wtICER (*n*=3). We note that Glycosaminoglycan biosynthesis of keratan sulfate, while the highest fold-enrichment, PI3K-Akt includes the lowest FDR, and includes many relevant genes involved in melanoma and other cancers including *akt1, cdkn1a, col4a4, fgf5, fgfr2, met, pdgfra, pdgfrb*. (B) Venn diagram illustrating the overlap of genes identified through analyses of ICER's transcriptional mechanisms. Total filtered list of 58 zebrafish and orthologous Human genes that were upregulated in wtICER, but downregulated in EGFP and phosphorylation mutant, and also contain CRE motifs in the promoter regions; 16 total genes overlap between 2 ChIP-Seq experiments from ENCODE (*crema, zcchc24, prkacaa, iqsec1b, rgs12b, sik2b, gli2a, gab1, ttyh3b, gyg1b, zbtb16b, actn4, junba, plxdc2, plpp3,* and *hmga2*).

N-terminal modification of ICER disrupts UBR4-mediated recognition, thereby preventing ubiquitination and degradation (Cirinelli et al., 2022). Additionally, phosphorylation of ICER at serine 35 by the mitotic kinase CDK1 triggers monoubiquitination, resulting in its cytosolic relocalization. To circumvent both degradation and relocalization, we engineered an ICER mutant with serine-to-alanine substitutions at positions 35 and 41 (S35-41A-ICER) and included an N-terminal HA-tag to prevent N-terminal ubiquitin ligase recognition. This ubiquitination-resistant ICER mutant resists both mono- and polyubiquitination, retaining its nuclear localization and stability.

In this study, we investigated the impact of the ubiquitination-resistant ICER using a *Danio rerio* (zebrafish) melanoma model expressing *braf* V600E, *p53* loss-of-function, and *mitf* loss-of-function, rescued through the miniCoopR system. This model restores *mitf* expression only upon successful transgene integration, providing a robust system to study tumorigenesis in melanocytes. Histological analysis (H&E staining) revealed striking differences in tumor invasiveness among zebrafish expressing HA-tagged wtICER, S35-41A mutant ICER, and an EGFP control (Fig. 3). The wtICER group exhibited the most aggressive tumor phenotype, while the S35-41A mutant ICER appears to have reduced tumor invasiveness. These findings suggest that the S35-41A ICER mutant sustains tumor-suppressive function, contrasting sharply with the highly invasive phenotype associated with wtICER. However, it is also possible that the wtICER has tumor promoting function, that is abrogated by the phosphorylation modifications After generating the cohorts and a 2-year observation period, we conducted a Kaplan–Meier survival analysis, which suggests a survival benefit conferred by the S35-41A mutant ICER. Zebrafish expressing this mutant exhibited a demonstrable prolonged survival compared to the wtICER or EGFP control. Indeed, the difference in lifespan between all groups via log-rank test was statistically significant (*P*<0.05) (Fig. 2). However, post-

hoc pairwise comparisons using *t*-tests, we observed significant differences between EGFP and wt-ICER, as well as between S35-41A ICER and wt-ICER, but not between EGFP and S35-41A ICER. Additional studies with increased sample size will be essential to improve statistical power and determine the impact of this mutation on disease progression and overall survival. Notably, the accelerated mortality observed in the wt-ICER group was associated with more aggressive tumor burden, supporting a model in which ICER instability contributes to tumor suppression. However, in light of these findings, an alternative hypothesis must also be considered: that overexpression of N-terminally unmodified ICER under the regulation of the *mitf* promoter, may exert tumor-promoting effects in this context, and that the S35-41A mutation, by altering phosphorylation-dependent regulation, rescues the phenotype toward a less pathogenic or more neutral state. Further investigation will be necessary to dissect whether these effects are due to loss of tumor-suppressive function, gain of oncogenic function, or altered regulation of ICER stability and activity *in vivo*. Alternatively, more work is needed to fully understand the expression stability of the S35-41A mutant ICER fish in this context, as we were unable to identify ICER protein via western blot in this study.

Our attempts to establish stable cell lines via the same protocol for the S35-41A mutant were unsuccessful. We hypothesize the inability to grow the cells derived from this phosphorylation deficient fish's neoplastic tissue is likely due to ICER's constitutive impact on the cell cycle, driven by its intentionally impaired post-translational regulation and underscores the influence of ubiquitination-resistant ICER on cellular proliferation and warrants further investigation.

We also explore the gene expression profiles of the wtICER and EGFP tumor derived cell lines, which revealed several differentially expressed genes, with many pathologically relevant pathways to explore. We note many of interest, including the upregulation of glycosaminoglycan keratan synthesis, as well as upregulated

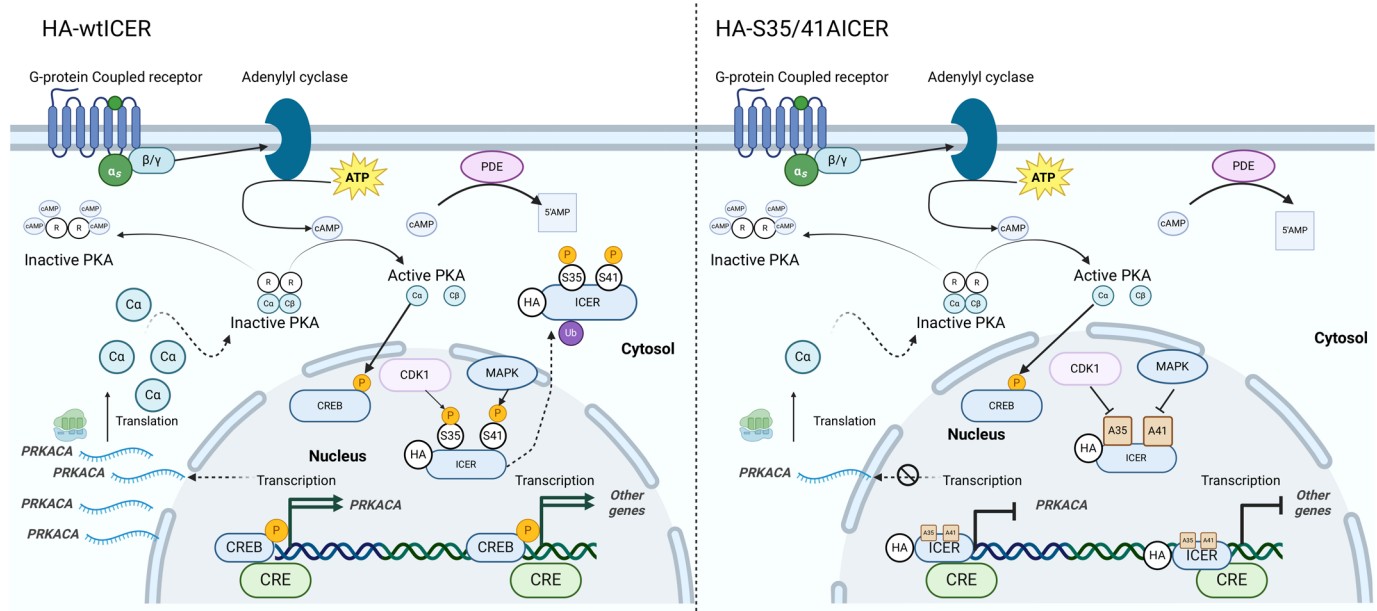

**Fig. 7. Proposed Model of PRKACA regulation via ICER.** Proposed model showing how wtICER (left) and the phosphorylation deficient S35-41-ICER (right) may impact gene expression, and specifically *PRKACA*, encoding the alpha catalytic subunit of PKA. Following adenylyl cyclase conversion of ATP to cAMP, 2 cAMP molecules bind to each PKA regulatory subunit, releasing the now active catalytic subunits, which phosphorylate CREB among other targets. Active CREB binds to CREs on the *PRKACA* promoter, upregulating gene expression of this subunit, which in turn, can more readily phosphorylate CREB in the presence of additional cAMP secondary messenger signal, forming a positive feedback loop. On the left, ICER is shown, in this case, to be phosphorylated on serine's 35 and 41, and subsequently ubiquitinated and removed to the cytosol. ICER in the example shown is not poly-ubiquitinated or degraded due to the presence of an HA-tag which prevents N-terminal recognition by UBR4. On the right, we show the phosphorylation deficient ICER is now unable to be phosphorylated, and therefore not ubiquitinated. In this environment, ICER competitively binds to CRE's on the promoter of many genes, including *PRKACA* and inhibits expression. In the case of *PRKACA* transcriptional repression by ICER, expression of the PKA catalytic subunit is decreased, decreasing the amount of phosphorylated and active CREB.

PI3K/AKT, Rap1 signaling, and upregulation of *gdf6*, which has been shown to be highly upregulated in melanoma (Ceol et al., 2011) however, the relationship, if any, between ICER and these pathways is not well understood. To begin to understand the mechanisms involved in promotion of a more aggressive melanoma in the zebrafish expressing wtICER, we focus here on an exploration of wtICER specific upregulated genes that have corresponding binding affinity for CREs, confirmed via ChIP-Seq data: *PRKACA* which encodes PKA catalytic α-subunit and glioma-associated oncogene homolog 2 (*GLI2*). *PRKACA* was of interest due to its demonstrated implication in MAPK pathway inhibitors through its restorative phosphorylation and activation of CREB, and because the region surrounding the TSS contained the highest number of CRE motifs, which includes half-CREs (TGACG/CGTCA) among the melanogenic relevant gene set via DAVID. Resistance to MAPK pathway inhibitors has been shown to involve GPCR-mediated signaling, where GPCR activation induces adenylyl cyclase, leading to the production of cAMP and subsequent activation of PKA. Supporting this, the adenyl cyclase gene, *ADCY9,* and the PKA catalytic subunit alpha (*PRKACA*) were identified as key resistance effectors in melanoma, with *PRKACA* displaying the highest rescue score among serine/threonine kinases in a gain-of-function resistance study (Johannessen et al., 2013). Both *ADCY9* and *PRKACA* were demonstrated to confer resistance across a range of MAPK pathway inhibitors. These findings highlight a signaling network involving cAMP–PKA signaling as a driver of drug resistance in melanoma. Downregulation of *PRKACA* in a murine model reduced CREB Serine-133 phosphorylation and inhibited many pro-proliferative pathways (Wang et al., 2022). To our knowledge, it has not previously been suggested that ICER, as a

transcriptional antagonist of CREB/CREM, is capable of inhibiting PKA-mediated phosphorylation of CREB, by downregulating *PRKACA,* the alpha catalytic subunit of PKA. The post-translational modifications of endogenous ICER, or the N-terminal HA-tagged ICER in this model, permit promoter CRE binding of CREB, which further stimulates PKA activity, and may partially explain the resurgence of phosphorylated CREB in drug-resistant melanoma. Our findings suggest that ICER may play a role in this signaling cascade by directly disrupting the cAMP–PKA–CREB feedback loop, by not only decreasing active CREB levels but also by mitigating the impact of the downstream targets of PKA. Here, we posit a novel mechanism (Fig. 7) through which ICER could counteract resistance to MAPK inhibitors and potentially restore tumor sensitivity to therapy, though we also suspect similar mechanisms across a constellation of regulatory elements by ICER and other transcription factors contribute to this mechanism to fully inhibit melanoma progression. Further investigation is warranted to explore ICER's regulatory role in this pathway and its therapeutic implications.

*GLI2*, an inducer of hedgehog (Hh) responsive genes, was of interest as another possible target of ICER regulation, because it also contains a high number of CRE motifs on the promoter region, and due to previous work highlighting the interest of this transcription factor in BRAF inhibitor (BRAFi) resistance. Notably, *in-vitro* melanoma cell lines with acquired BRAFi resistance show increased expression of *GLI2* and that knockdown of *GLI2*, restores sensitivity to vemurafenib (Faião-Flores et al., 2017). In other experiments, reduction in *GLI2* expression, inhibited both basal and Transforming Growth Factor-β (TGF-β) induced cell migration via matrigel assay, suggesting that the upregulation of *GLI2* contributes to a more

invasive phenotype (Alexaki et al., 2010; Faião-Flores et al., 2017). *GLI2* upregulation in melanoma was also associated with mesenchymal transition, which promotes cell-cell adhesion and the other motile behavior and is typically associated with poor prognosis (Javelaud et al., 2011). Interestingly, *gli2a* induced upregulated genes, such as *gli1* and *ccnd1*, were not identified as differentially expressed in our RNA-seq data. We hypothesize that while CREB/CREM can bind to motifs on the *gli2a* promoter and enhance its transcription (and ICER inhibits this process), the cells remain in a Hh-inactive state. In this state, PKA phosphorylates gli2a, leading to proteasomal degradation of the protein (Li et al., 2017). This degradation leaves only the transcriptional repressor domain active, which, in the absence of Hh signaling, causes no significant change in transcriptional activity.

Collectively, our results indicate that the degradation of wtICER disrupts its tumor-suppressive function, accelerating melanoma progression in this zebrafish model. In contrast, the ubiquitin-resistant S35-41A ICER mutant retains tumor-suppressive activity, evidenced by decreased tumor invasiveness and improved survival. We show that the autochthonous wtICER expression in melanocytes in a *braf*^V600E, *p53*(lof) context leads to upregulation of metastatic-promoting genes involved in *PI3K/AKT/RAP1* signaling and glycosaminoglycan keratan sulfate synthesis, as well as progression towards a less differentiated melanocyte phenotype. Additionally, we observe increased expression of key regulators of cell signaling and division, including *pdgfra*, *met*, and *notch3*. However, a direct mechanistic link between these pathways and ICER remains unclear, raising the possibility that resistant cells are being selectively enriched, or that additional, as-yet-uncharacterized mechanisms are at play. We offer some insight into the consequence of a ubiquitin deficient ICER, and what genes/pathways may be selected for in response to this environment, but more work will be needed to elucidate how a ubiquitin-deficient ICER can inhibit tumorigenesis.

## MATERIALS AND METHODS
### miniCoopR vector and ICER constructs
To investigate the effects of candidate genes on melanoma progression, we used the *miniCoopR* vector system, a transposon-based tool specifically designed for transgenic zebrafish models, as described in Ceol et al. (2011) and Iyengar et al. (2012). The *miniCoopR* vector contains a *mitfa* rescuing minigene, which includes the *mitfa* promoter, open reading frame, and 3′-untranslated region of the wild-type *mitfa* gene (Addgene #118850). This rescuing element enables melanocyte development in *Tg(mitfa:BRAFV600E); p53(lf);mitfa(lf)* zebrafish, where *mitfa* loss-of-function *mitfa*(lf) otherwise blocks melanocyte formation. Candidate melanoma-modifying genes were cloned into the *miniCoopR* vector using Gateway cloning. The resulting constructs were microinjected into single-cell *Tg(mitfa:BRAFV600E);p53(lf); mitfa(lf)* zebrafish embryos alongside *Tol2* transposase RNA to facilitate stable genomic integration. Successful *miniCoopR* integration not only rescues melanocyte development but also drives candidate gene expression within these melanocytes. For this study, we used the *miniCoopR* system to express two ICER cDNA variants: wild-type ICER-Iγ (wtICER) and a phosphorylation-resistant ICER-Iγ mutant (ICER-S35&41A). Additionally, the *miniCoopR* plasmid contained the *EGFP* transgene, as previously described. Both ICER constructs were synthesized by Genescript USA Inc. and cloned into *pENTR™/D-TOPO®* (Invitrogen) before being transferred into the *miniCoopR* vector using the MultiSite Gateway® System (Invitrogen). Successful cloning was confirmed by sequencing, and the final constructs were prepared for zebrafish embryo injections.

### DNA construct generation of mutant ICER plasmids for gateway cloning
Platinum™ SuperFi™ DNA Polymerase was used to amplify the mutant ICER constructs from mutant ICER plasmids. A CACC sequence was added to the 5′ end of the forward primer to create a PCR product suitable for a TOPO reaction. The blunt ends were prepared using Platinum SuperFi Polymerase. A TOPO reaction was then performed with the ICER PCR product and TOPO vector, generating a plasmid with attL1 and attL2 sites flanking the ICER sequence for Gateway cloning. An LR clonase reaction followed, using the ICER TOPO plasmid, miniCoopR plasmid, mitfa promoter plasmid, and poly-A signal plasmid to create the final construct.

### Zebrafish breeding, embryo collection, and microinjection
#### Breeding setup
Zebrafish were maintained on a 14-h light:10-h dark cycle. The night before collection, breeding tanks were set up, simulating natural mating environments. Males and females were placed on opposite sides of a divider, and mating commenced in the morning. Freshly fertilized embryos were collected, washed, and stored in methylene blue water.

#### Microinjection protocol
Embryos were cooled to 4°C to arrest cellular division and then transferred to 18°C for microinjection. Injection material, comprising Tol2 RNA, the DNA construct, and a phenol red indicator, was injected into embryos at the one-cell stage. These eggs were incubated at 28°C, and melanin production was assessed to confirm successful injection.

Injected embryos were monitored until hatching (typically 3 days post-fertilization). Embryos with successful *mitf* rescue (evidenced by melanin production) were candidates for the survival study, though only fish that developed macroscopic lesions were included in the survival study.

### Kaplan–Meier survival study
Kaplan–Meier survival analysis was performed using the *survfit()* function in R, with overall survival differences between groups initially assessed using the log-rank test via the *survdiff()* function ($\alpha=0.05$). Log-Rank (Mantel-Cox) test indicated a statistically significant difference in overall survival among the three (EGFP, wt-ICER, and S35-41A-ICER) mutant zebrafish groups ($P=0.0004$), leading us to reject the null hypothesis that survival is equal across all groups. Despite this significance, the Kaplan–Meier plot revealed similar median survival times between the EGFP and S35-41A-ICER groups. Consistent with this observation, the pair-wise *t*-test showed no statistically significant difference in overall survival between EGFP and S35-41A-ICER ($P>0.05$).

### Western blotting and histology
Sodium dodecyl sulfate polyacrylamide gel electrophoresis (SDS-PAGE) and western blot analysis for ICER expression was performed as previously described (Healey et al., 2013). The anti-ICER polyclonal antibody was prepared from rabbits immunized against ICER protein as previously described in Molina, et al. (1993). Anti α-tubulin antibody (polyclonal) was purchased from Sigma-Aldrich. WB detection was performed using IRDye® secondary antibodies and imaged using the Odyssey® CLx infrared system from LI-COR. Western blot analysis using anti ICER antibodies were performed as described before (Healey et al., 2013). Western blot detection was performed using IRDye® secondary antibodies and imaged using the Odyssey® CLx infrared system from LI-COR. Tubulin antibodies (TU-02) was purchased from Santa Cruz Biotechnology and used at a 1:500 dilution for Western blotting.

To assess tumor invasiveness, tumors were collected 2 weeks after the first appearance of a lesion for histological review. Sections were reviewed for evidence of muscle invasion, defined as tumor cell infiltration beyond the dermal boundary into adjacent skeletal muscle fibers, as judged by disruption of the dermal-muscular interface and cellular infiltration patterns. HA-wtICER group: five out of six fish showed clear muscle invasion. EGFP control group: none of four evaluable tumors showed invasion. HA-S35&41A ICER mutant group: Tumors from more than six independent fish were evaluated across replicate experiments, and none showed evidence of muscle invasion. While only one high-resolution representative image is included in the main figure, this image reflects the best histological section available. All findings were confirmed across multiple independent samples.

## FACS mitfa+ isolation

Skin was manually dissected from *mitfa:mCherry⁺; crestin:EGFP⁺; tyrosinase⁻/⁻* zebrafish from two cohorts (at 6-or 14 months post fertilization) and finely chopped for 1–2 min. Tissue was digested in 1 ml TrypLE Express (ThermoFisher Scientific, 12605028) at 37°C with shaking at 300 rpm for 45 min. The resulting cell suspension was filtered through a 100 µm cell strainer (ThermoFisher Scientific, 22-363-549) and washed with 5 ml of FACS buffer consisting of DPBS (ThermoFisher Scientific, 14190144), 1% Penicillin-Streptomycin (ThermoFisher Scientific, 15140122), and 2% heat-inactivated FBS (ThermoFisher Scientific, A3840001). Cells were centrifuged at 500× *g* for 5 min, resuspended in 200 µl FACS buffer, and stained with DRAQ7 viability dye (Abcam, ab109202) immediately prior to sorting. Live melanocytes (DRAQ7⁻, mCherry⁺) were isolated using a FACSAria cell sorter (BD Biosciences) with a 100 µm, 20 PSI nozzle. 98,000 sorted melanocytes and mitfa+ cells were pooled from five to seven zebrafish and used for downstream lysate preparation for western blotting.

## Sorted mitfa+ Cell lysate preparation

Sorted cells were lysed in radio-immunoprecipitation assay (RIPA) Lysis and Extraction Buffer (ThermoFisher Scientific, 89900) supplemented with protease inhibitor cocktail (ThermoFisher Scientific, A32955) and triturated with a U-100 insulin syringe. Lysates were centrifuged at 13,300 rpm for 20 min at 4°C. The supernatants were collected and stored at −80°C.

## Tumor disaggregation and cell isolation

Zebrafish were euthanized on ice, and tumors were surgically excised using a scalpel in a Petri dish. Tumors were transferred into a small Petri dish containing 2 ml of dissection medium, which consisted of 50% Ham's F12, 50% DMEM, 10× Pen/Strep, and 0.075 mg/ml Liberase. Tumor samples were incubated in the dissection medium for 30 min at room temperature before manually disaggregating the tumor with a scalpel. The cell suspension was then filtered through a 40 µm cell strainer, and residual tumor chunks were further disrupted. Cells were washed with inactivation/wash medium, composed of 50% Ham's F12, 50% DMEM, 10× Pen/Strep, and 15% FCS. After centrifugation at 500 rcf for 5 min, cells were resuspended in zebrafish cell medium.

## Fibronectin coating

We coated the appropriate wells of cell culture plates with fibronectin at a concentration of 10 µg/ml, prepared in sterile cell culture PBS, for 30 min at room temperature. The wells were then coated with the following volumes based on plate type: 250 µl per well for 48-well plates, 375 µl for 24-well plates, 500 µl for 12-well plates, and 1 ml for six-well plates. Following incubation, we collected and stored the fibronectin solution at 4°C for reuse within 2-3 weeks.

## Cell culture and splitting

Cells were plated on fibronectin-coated 48-well plates in 500 µl of zebrafish cell medium, which was composed of a 1:1 mixture of DMEM and Ham's F12 with added glutamine, 15% FBS, 10% zebrafish embryo extract, insulin, holotransferrin, selenous acid, chemically defined lipids, non-essential amino acids, and other essential supplements. For splitting cells, we washed with sterile PBS, and trypsinized cells for 2–5 min at 37°C. After trypsinization, the reaction was inactivated with 10% serum-containing medium, and cells were transferred to progressively larger plates as needed.

## RNA extraction, sequencing and analysis

For cells, 10⁶ cells were pelleted prior to total RNA extraction, solid tumors excised from the fish varied by mass (20-40mg) and volume of RNA lysis buffer was added according following the NEB #T2010 kit from New England BioScience. RNA-seq libraries were then prepared using the NEBNext Ultra II DNA Library Prep Kit for Illumina (NEB# E7645), and rRNA depletion was achieved with the QIAgen QIAseq FastSelect rRNA Fish Kit (333252). The final RNA libraries were pooled in an equimolar fashion and paired-end sequenced on an Illumina platform with read lengths of 151bp. Fastq files were trimmed for Illumina adapters using Trim Galore! version 0.6.7 and aligned to zebrafish GRCz11 via BWA-MEM2 version 2.2.1. Read count was performed using HTseq (version 2.0.5) (Putri et al., 2022). Read counts were combined and differential gene expression analysis was conducted using iDEP (version 2.0.1) via DESeq2, with FDR ≤0.05, and log2 (fold-change) ≥2 (Ge et al., 2018). GO was conducted using ShinyGO with an FDR of ≤0.05 (Ge et al., 2020).

## Gene set filtering

DEG genes analyzed via iDEP were filtered for those that were upregulated, in the wtICER tumor derived cell line, relative to the EGFP tumor derived cell line, and upstream regions flanking gene were obtained using ENSEMBL Biomart. Exported upstream regions were analyzed using FIMO via the MEME-suite version 5.5.7 with $P<0.001$ cutoff searching for full canonical CREs (5′ TGACGTCA 3′) and half-CREs (5′ TGACG 3′/5′ CGTCA 3′) (Grant et al., 2011). Genes containing these motifs, or versions within the algorithmic limit of differentiation, were exported and genes were compared to solid tumor bulk-RNA seq read counts, after normalization based on number of reads, and filtered for those in which were downregulated by at least 2-fold in the phosphorylation mutant ICER. Upstream regions were then exported for the orthologous hg38 human gene promoter region, and confirmed for CRE type motif as in the cell analysis. ENCODE project Transcription factor ChIP-Seq data was then analyzed using the output narrow peak bed files of experiment ENCSR903ELW (HepG2 biosample) and ENCSR077DKV (K562 biosample). Coordinates were converted to gene IDs and gene symbols via UCSC table browser and compared to filtered RNAseq candidate genes of interest.

## Acknowledgements

We want to thank Drs Leonard Zon, Charles Kaufmans, Maurizio Fazio, Isabelle Roszko for helping establish our zebrafish facility, help with embryo injection techniques and general pathological analysis. Drs Julien Ablain and Richard White for helping establish the fish melanoma cell lines. Dr Laying Wu for tissue processing and microscopy. Rachel Hongo and Boston Children's Hospital HEM/ONC-HSCI Flow Cytometry Research Laboratory for performing melanocytes isolation from zebrafish via FACS.

## Competing interests

The authors declare no competing or financial interests.

## Author contributions

Conceptualization: J.W., C.A.M.; Data curation: J.W.; Formal analysis: J.W.; Funding acquisition: C.A.M.; Investigation: J.W., M.S., A.C., J.R.; Methodology: J.W., A.C., C.A.M.; Writing – original draft: J.W.; Writing – review & editing: C.A.M.

## Funding

This work was supported by the National Institutes of Health [grant number 5SC1GM125583]. Open Access funding provided by College of Science and Mathematics and Biology Department, Montclair State University, Montclair New Jersey, USA 07043. Deposited in PMC for immediate release.

## Ethics statement

Zebrafish were maintained in accordance with the guidelines set forth by the Institutional Animal Care and Use Committee (IACUC) to ensure ethical and humane treatment of vertebrate animals in research. All experiments involving zebrafish were carefully designed to minimize pain and distress, using appropriate anesthetics and analgesics when necessary. Housing and care comply with the Guide for the Care and Use of Laboratory Animals, including maintaining zebrafish in well-regulated aquatic facilities with optimal water quality and enrichment. Protocols undergo rigorous review and approval by the IACUC, ensuring that the principles of the 3Rs – Replacement, Reduction, and Refinement – were upheld throughout our research endeavors.

## Data and resource availability

All relevant data can be found within the article and its supplementary information.

## Peer review history

The peer review history is available online at https://journals.biologists.com/bio/lookup/doi/10.1242/bio.061904.reviewer-comments.pdf

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
