## [Peer Review File · Biology Open]

Phosphorylation Deficient Inducible cAMP Early Repressor (ICER) Modulates Tumorigenesis and Survival in a Transgenic Zebrafish (*Danio rerio*) Model of Melanoma

Melissa Spigelman, Angelo Cirinelli, James Reilly, Carlos A. Molina and Justin Wheelan
DOI: 10.1242/bio.061904

Editor: Christopher A. Maher

Review timeline

Original submission:	22 January 2025
Editorial decision:	18 February 2025
First revision received:	19 May 2025
Editorial decision:	9 June 2025
Second revision received:	27 June 2025
Accepted:	14 July 2025

Original submission

First decision letter

MS ID#: bio.061904

MS TITLE: Phosphorylation Deficient Inducible cAMP Early Repressor (ICER) Modulates Tumorigenesis and Survival in a Transgenic Zebrafish (*Danio rerio*) Model of Melanoma

AUTHORS: Justin Wheelan; Melissa Spigelman; Angelo Cirinelli; James Reilly; Carlos A. Molina

I have now reached a decision on the above manuscript.

The reviewer reports are shown at the bottom of this email or can be accessed, together with a copy of this decision letter, by going to:

As you will see, the reviewers raised a number of substantial criticisms that prevent me from accepting the paper at this stage.

They suggest, however, that a revised version might prove acceptable, if you can address their concerns. If you think that you can deal satisfactorily with the criticisms on revision, I would be pleased to see a revised manuscript. We would then return it to the reviewers.

At this stage, we also ask you to ensure your manuscript complies with our formatting guidelines. Provided you are able to fully address the referees' comments, we are positive about publication of your paper (we accept over 95% of revision submissions) and therefore hope you won't mind any extra work involved in reformatting your manuscript at this point.

Please ensure that you clearly highlight all changes made in the revised manuscript. Please avoid using 'Tracked changes' in Word files as these are lost in PDF conversion.

I should be grateful if you would also provide a point-by-point response detailing how you have dealt with the points raised by the reviewers in the 'Response to Reviewers' box. Please attend to

all of the reviewers' comments. If you do not agree with any of their criticisms or suggestions please explain clearly why this is so.

Reviewer 1

Comments for the author

Melanoma has been commonly associated with mutations in the BRAF gene which have been shown to increase tumor proliferation by means of activating the ERK1/2 signaling cascade. Though these mutations have provided information on melanoma tumor progression, treatment via BRAF inhibitors has encountered significant challenges with therapeutic resistance. Resistance to BRAF inhibitors in melanoma has been linked to the activation of cAMP-dependent signaling networks involving protein kinase A (PKA) and CREB. The manuscript by Wheelan et al. focuses on the Inducible cAMP early repressor, ICER (an antagonist of CREB), and its potential role in tumor progression. Specifically, phosphorylation at serines 35 and 41 (followed by ubiquitination) marks ICER for proteasomal degradation, and this manuscript investigates the role of S35 and S41 in melanoma. As the study aims to identify the impact of ICER phosphorylation on melanoma growth, they employed the zebrafish miniCoopR system to generate three ICER transgenic cohorts to observe over 104 weeks. Further pathology staining, PCR, and western blot analysis were utilized to support their conclusions about S35-41A mutant ICER sustaining tumor suppression. While their conclusions may shed some light on ICER and cAMP biology in melanoma, the authors must take care with their interpretations of data and conclusions drawn from these interpretations.

Major Comments:

1. Figure 2. Kaplan-Meier Survival Curve:

- a. Analysis of the results displayed in Figure 2 suggest that wt-ICER overexpression has a negative impact on zebrafish survival. To support this claim that a lack of similar properties with the S35-41A-ICER is due to the ablation of phosphorylation sites, it would be beneficial to provide evidence that the S35-41A-ICER protein is both stable and detectable within tumors.
- b. The survival analysis curve may include fish death that was unrelated to induced melanomas (e.g. MPNST tumors caused by loss of p53). It would strengthen the findings if the data presented are exclusively limited to fish that developed melanoma.

2. Figure 3. Tumor Histology and Phenotypic Analysis:

- a. The images provided of the melanoma and invasive melanoma zebrafish appear to be misrepresentative of their corresponding pathology stains. We would expect that the miniCoopR-EGFP zebrafish, which have a lesion that appears to have skin and scales over it, to show an invasive pathology, and the miniCoopR-wt-HAICER, which have a lesion growing over skin and scales, to show a less or non-invasive pathology. Is there an explanation for why the nature of the tumors and the presented zebrafish phenotypes appear inconsistent with one another?
- b. Although difficult to discern from the image provided, it appears that the pathological staining for miniCoopR-HAS35/41A-ICER - described as a "benign lesion" - appears to show a great deal of cellularity that is consistent with it being a tumor. The authors should provide clearer images, and, if this lesion is benign, provide criteria for defining it as such.

3. The authors should modify the interpretations of their data and conclusions to ensure that their analysis is fully comprehensive and consistent with their findings. Below are a few examples in which the authors generalize their results:

- a. The findings in Figure 1 are described in a manner that claims the results suggest that ICER is abnormally degraded in tumor tissues. However, the figure provided along with the supporting arguments in the section titled "Expression of ICER protein is abnormal in zebrafish melanomas" do not provide sufficient evidence to draw this conclusion. The western blot does not directly look at ICER degradation, and the authors should be careful not to exclude other possibilities such as transcriptional downregulation or transgene silencing.

b. In the Discussion, the authors claim that ICER has a tumor-suppressive function and that phosphorylation and degradation of ICER disrupts this function. If this were the case, then one would expect S35-41 ICER overexpression to suppress tumor formation relative to control animals. Instead, there is no difference between the control and S35-41 ICER survival curves. They do see that wtICER overexpression decreases survival relative to control and S35-41 ICER animals. The simplest interpretation would seemingly be that wtICER has a tumor promoting function and that the S35-41 alterations abolish this function. The authors either need to adopt this interpretation or explain why they do not adopt it.

Minor Comments:

1. The western blot analysis in Figure 1 compares skin tissues to melanomas. The cell types in the skin samples include fibroblasts, keratinocytes, and a small portion of melanocytes. By contrast, melanomas are composed mostly of melanocytic cells. It is possible that expression of ICER in other skin cell types is higher than in melanocytes or melanoma cells. The authors should acknowledge that this is another interpretation of their results.

2. The use of the Student's t test in Figure 5 is not sufficiently described. Were single time points compared, and, if so, which ones were significantly different?

Reviewer 2

Comments for the author

This study explores the role of Inducible cAMP Early Repressor (ICER) in melanoma progression, particularly in the context of BRAFV600E-driven melanoma using a zebrafish melanoma model. The researchers highlight that ICER, a transcriptional repressor, is degraded in melanoma, undermining its tumor-suppressive effects. Using a BRAFV600E; p53(loss of function) transgenic zebrafish model, they tested a ubiquitin-resistant ICER mutant and found that it prolonged survival and reduced tumor invasiveness compared to the wild-type ICER. RNA sequencing revealed that the wild-type ICER altered the expression of key oncogenic pathways, including CREB/CREM and PI3K/AKT signaling. These findings suggest that stabilizing ICER may provide a potential therapeutic strategy for suppressing melanoma progression and overcoming drug resistance in BRAFV600E-mutant melanoma. This study is intriguing, but the data currently provided is not sufficient to support some of the principal conclusions of this study.

Major points:

1. Fig 1: The authors conclude that ICER protein is downregulated in melanoma tumors based on their comparison with normal skin but not melanocytes. Have the authors detected ICER expression in melanocytes only? This is a critical control to support this conclusion.

2. Fig 3: The histology data is not convincing in its current form. This reviewer suggests providing higher resolution images of each tumor where the corresponding muscle tissue is clearly visible in each tumor. The reviewer further encourages the authors to provide quantifications of the invasive phenotype with numbers for each condition to conclusively demonstrate how many fish per tumor condition show the stated phenotype. The current representative images are difficult to interpret and need further clarification.

Minor points:

1. The differentially expressed genes are interesting (Fig. 6) but there is no information about the statistical significance of these findings in the main text or the methods section. This reviewer encourages the authors to provide this information in detail with the exact parameters used in iDEP to obtain differentially expressed genes.

Reviewer's Responses to Questions

Experimental quality

Does each figure have the proper controls?

If 'No', please indicate reasons in Comments for Author box below.

Reviewer #1:

- Yes

Reviewer #2:

- Yes

Were the data analyzed using appropriate statistical tests?

If 'No', please indicate reasons in Comments for Author box below.

Reviewer #1:

- Yes

Reviewer #2:

- Yes

Reproducibility

Were experiments performed using adequate number of biological replicates?

If 'No', please indicate reasons in Comments for Author box below.

Reviewer #1:

- Yes

Reviewer #2:

- Yes

Does the methods section provide sufficient detail to permit reproducibility?

If 'No', please indicate reasons in Comments for Author box below.

Reviewer #1:

- Yes

Reviewer #2:

- Yes

Completeness

Are the manuscript's conclusions supported by the data?

If 'No', please indicate reasons in Comments for Author box below.

Reviewer #1:

- No

Reviewer #2:

- No

Scholarship

Do the authors cite and discuss the merits of data that would argue for and against their conclusion?

If 'No', please indicate reasons in Comments for Author box below.

Reviewer #1:

- Yes

Reviewer #2:

- Yes

Does the manuscript title & abstract accurately reflect the contents of the manuscript, without hyperbole?

If 'No', please indicate reasons in Comments for Author box below.

Reviewer #1:

- No

Reviewer #2:

- Yes

First revision

Author response to reviewers' comments

Comments from the Reviewers:

Reviewer 1: Melanoma has been commonly associated with mutations in the BRAF gene which have been shown to increase tumor proliferation by means of activating the ERK1/2 signaling cascade. Though these mutations have provided information on melanoma tumor progression, treatment via BRAF inhibitors has encountered significant challenges with therapeutic resistance. Resistance to BRAF inhibitors in melanoma has been linked to the activation of cAMP-dependent signaling networks involving protein kinase A (PKA) and CREB. The manuscript by Wheelan et al. focuses on the Inducible cAMP early repressor, ICER (an antagonist of CREB), and its potential role in tumor progression. Specifically, phosphorylation at serines 35 and 41 (followed by ubiquitination) marks ICER for proteasomal degradation, and this manuscript investigates the role of S35 and S41 in melanoma. As the study aims to identify the impact of ICER phosphorylation on melanoma growth, they employed the zebrafish miniCoopR system to generate three ICER transgenic cohorts to observe over 104 weeks. Further pathology staining, PCR, and western blot analysis were utilized to support their conclusions about S35-41A mutant ICER sustaining tumor suppression. While their conclusions may shed some light on ICER and cAMP biology in melanoma, the authors must take care with their interpretations of data and conclusions drawn from these interpretations.

Major Comments:

1. Figure 2. Kaplan-Meier Survival Curve:

a. Analysis of the results displayed in Figure 2 suggest that wt-ICER overexpression has a negative impact on zebrafish survival. To support this claim that a lack of similar properties with the S35-41A-ICER is due to the ablation of phosphorylation sites, it would be beneficial to provide evidence that the S35-41A-ICER protein is both stable and detectable within tumors.

We have attempted to identify S35&S41A-ICER via Western blot in 20 separate experiments, however we were unable to identify ICER protein from any of the fish in this cohort. We also do not observe eumelanin via gross observation of the fish around the location of what we believed was a benign lesion. It seems likely that given the p53 loss-of-function in these fish, they likely have developed MPNST as suggested, which we have adopted into our interpretation of the fish pathology.

b. The survival analysis curve may include fish death that was unrelated to induced melanomas (e.g. MPNST tumors caused by loss of p53). It would strengthen the findings if the data presented are exclusively limited to fish that developed melanoma.

Methodology in the experiment did select for only fish with tumors, however it was not possible in the scope of this study to determine if fish developed other tumor types as pathology was only performed in a subset of samples. Method section updated to include more detail into this process, as fish that developed macroscopic lesions were included in the analysis. Results section updated to reflect that while we do not know for sure that each fish indeed had melanoma, only fish developing lesions were counted in the study. We still find it useful to include those who may have had similar presenting lesions, though as suggested, indeed it was possible that some of the fish in the study did indeed develop MPNST, and we appreciate this observation.

2. Figure 3. Tumor Histology and Phenotypic Analysis:

a. The images provided of the melanoma and invasive melanoma zebrafish appear to be misrepresentative of their corresponding pathology stains. We would expect that the miniCoopR-EGFP zebrafish, which have a lesion that appears to have skin and scales over it, to show an invasive pathology, and the miniCoopR-wt-HAICER, which have a lesion growing over skin and scales, to show a less or non-invasive pathology. Is there an explanation for why the nature of the tumors and the presented zebrafish phenotypes appear inconsistent with one another?

Indeed, this is also what the Author's hypothesized would happen but was consistently not the case. The phenotype was representative of the wt-HAICER expressing cohort. We do not fully understand the mechanism, however we do attempt to connect the dots from the histology to the diminished survival and upregulation of pro-oncogenic genes that we hypothesize are contributing to the increase in lesion appearance. To the reviewer's point, we removed the descriptions on Figure 3 regarding the fish pathology.

b. Although difficult to discern from the image provided, it appears that the pathological staining for miniCoopR-HAS35/41A-ICER - described as a "benign lesion" - appears to show a great deal of cellularity that is consistent with it being a tumor. The authors should provide clearer images, and, if this lesion is benign, provide criteria for defining it as such.

Based on the reviewer's previous observation, we appreciate the comment and acknowledge we do not have a formal methodology to determine if this is a benign lesion, or if the fish are developing other non-melanoma tumors as a result of their genetic makeup. This is also the best representative images we have of the results. To that end, we removed the labels in Figure. 3 labelling the lesion as benign.

3. The authors should modify the interpretations of their data and conclusions to ensure that their analysis is fully comprehensive and consistent with their findings. Below are a few examples in which the authors generalize their results:

a. The findings in Figure 1 are described in a manner that claims the results suggest that ICER is abnormally degraded in tumor tissues. However, the figure provided along with the supporting arguments in the section titled "Expression of ICER protein is abnormal in zebrafish melanomas" do not provide sufficient evidence to draw this conclusion. The western blot does not directly look at ICER degradation, and the authors should be careful not to exclude other possibilities such as transcriptional downregulation or transgene silencing.

We appreciate the feedback, we included additional data to support this claim including Fig 1B, which supports previous observations by our group and others regarding ICER expression in melanoma. However, it is correct that in this context, we do not have direct evidence regarding post-translational modification in our current model, though we have shown ICER to be regulated post-translationally in other models, we acknowledge there are other possibilities, which we discuss in this section with the language that the mechanism of action is still under active investigation.

b. In the Discussion, the authors claim that ICER has a tumor-suppressive function and that phosphorylation and degradation of ICER disrupts this function. If this were the case, then one would expect S35-41 ICER overexpression to suppress tumor formation relative to control animals. Instead, there is no difference between the control and S35-41 ICER survival curves. They do see that wtICER overexpression decreases survival relative to control and S35-41 ICER animals. The simplest interpretation would seemingly be that wtICER has a tumor promoting function and that the S35-41 alterations abolish this function. The authors either need to adopt this interpretation or explain why they do not adopt it.

We appreciate the feedback. We included this interpretation as an alternative explanation, as the exact mechanism of action is still under investigation.

Minor Comments:

1. The western blot analysis in Figure 1 compares skin tissues to melanomas. The cell types in the skin samples include fibroblasts, keratinocytes, and a small portion of melanocytes. By contrast, melanomas are composed mostly of melanocytic cells. It is possible that expression of ICER in other skin cell types is higher than in melanocytes or melanoma cells. The authors should acknowledge that this is another interpretation of their results.

See comments above, we included an additional Western blot in Figure 1 that addresses this concern by assaying melanocytes directly as well as the tumor.

2. The use of the Student's t test in Figure 5 is not sufficiently described. Were single time points compared, and, if so, which ones were significantly different?

Upon further review by the author's, the figure this comment pertains to was deemed not to contribute meaningfully to the overall findings and was therefore removed.

Reviewer 2: This study explores the role of Inducible cAMP Early Repressor (ICER) in melanoma progression, particularly in the context of BRAFV600E-driven melanoma using a zebrafish melanoma model. The researchers highlight that ICER, a transcriptional repressor, is degraded in melanoma, undermining its tumor-suppressive effects. Using a BRAFV600E; p53(loss of function) transgenic zebrafish model, they tested a ubiquitin-resistant ICER mutant and found that it prolonged survival and reduced tumor invasiveness compared to the wild-type ICER. RNA sequencing revealed that the wild-type ICER altered the expression of key oncogenic pathways, including CREB/CREM and PI3K/AKT signaling. These findings suggest that stabilizing ICER may provide a potential therapeutic strategy for suppressing melanoma progression and overcoming drug resistance in BRAFV600E-mutant melanoma. This study is intriguing, but the data currently provided is not sufficient to support some of the principal conclusions of this study.

Major points:

1. Fig 1: The authors conclude that ICER protein is downregulated in melanoma tumors based on their comparison with normal skin but not melanocytes. Have the authors detected ICER expression in melanocytes only? This is a critical control to support this conclusion.

We appreciate the feedback, we included additional data to support this claim including Fig 1B, which supports previous observations by our group and others regarding ICER expression in melanoma compared with melanocytes only.

2. Fig 3: The histology data is not convincing in its current form. This reviewer suggests providing higher resolution images of each tumor where the corresponding muscle tissue is clearly visible in each tumor. The reviewer further encourages the authors to provide quantifications of the invasive phenotype with numbers for each condition to conclusively demonstrate how many fish per tumor condition show the stated phenotype. The current representative images are difficult to interpret and need further clarification.

We thank the reviewer for this helpful suggestion. We acknowledge that the histological images presented in the current figure may not provide optimal clarity, particularly in relation to muscle invasion. Unfortunately, the image shown in Fig. 3 is the highest quality section available for visualizing this phenotype, and we were limited by the tissue integrity and section orientation in some samples. However, we have now clarified the criteria used to assess the invasive phenotype based on the histopathological review.

Specifically, we observed that 5 out of 6 fish expressing HA-tagged wild-type ICER (HA-wtICER) exhibited histological evidence of tumor invasion into underlying skeletal muscle, as defined by loss of tissue boundary and infiltration of tumor cells into the muscle layer. In contrast, none of the 4 EGFP control fish with available interpretable sections showed any evidence of muscle invasion. For the HA-tagged S35&41A ICER mutant, more than 6 independent tumors were examined across multiple experiments, and none demonstrated evidence of invasion into muscle tissue. All tumors were collected at a standardized timepoint, two weeks following initial tumor appearance. We have now clarified these findings in both the main text and figure legend.

Minor points:

1. The differentially expressed genes are interesting (Fig. 6) but there is no information about the statistical significance of these findings in the main text or the methods section. This reviewer encourages the authors to provide this information in detail with the exact parameters used in iDEP to obtain differentially expressed genes.

We appreciate the feedback and have made this clearer by including the settings in the results as well as the methods section. Parameters set in iDEP to obtain differentially expressed genes were Read counts were combined and differential gene expression analysis was conducted using iDEP (v.2.0.1) via DESeq2 with FDR ≤ 0.05 , and \log_2 (fold-change) ≥ 2 (Ge et al., 2018). Gene Ontology was conducted using ShinyGO with an FDR of ≤ 0.05 (Ge et al., 2020).

Second decision letter

MS ID#: bio.061904R1

MS TITLE: Phosphorylation Deficient Inducible cAMP Early Repressor (ICER) Modulates Tumorigenesis and Survival in a Transgenic Zebrafish (*Danio rerio*) Model of Melanoma

AUTHORS: Justin Wheelan; Melissa Spigelman; Angelo Cirinelli; James Reilly; Carlos A. Molina

I have now reached a decision on the above manuscript.

The reviewer reports are shown at the bottom of this email or can be accessed, together with a copy of this decision letter, by going to:

As you will see, the reviewers gave favourable reports, but raised some critical points that will require amendments to your manuscript. I hope that you will be able to carry these out, because we would like to be able to accept your paper.

At this stage, we also ask you to ensure your manuscript complies with our formatting guidelines “ please see our manuscript preparation guidelines for details. Provided you are able to fully address the referees’ comments, we are positive about publication of your paper (we accept over 95% of revision submissions) and therefore hope you won’t mind any extra work involved in reformatting your manuscript at this point.

Please ensure that you clearly highlight all changes made in the revised manuscript. Please avoid using 'Tracked changes' in Word files as these are lost in PDF conversion.

I should be grateful if you would also provide a point-by-point response detailing how you have dealt with the points raised by the reviewers in the 'Response to Reviewers' box. Please attend to all of the reviewers’ comments. If you do not agree with any of their criticisms or suggestions please explain clearly why this is so.

Reviewer 1

Comments for the author

Major comment 1a: The authors do not directly address the possibility that S35-41A-ICER is not stably expressed; they should do so in the revised manuscript. Additionally, there is a discrepancy describing statistical significance in the results and discussion sections. In page 8 (line165) results the authors say that the difference between S35-41A ICER and EGFP lifespan was not statistically significant. However, in their discussion (page 8, line 333), authors claim that lifespan differences between all groups were statistically significant. The authors need to make this consistent.

Major comment 1b: The revised text pertaining to this comment needs clarification. It currently reads as though only animals with lesions were considered in the survival analysis. It should be clarified that all animals with rescued melanocytes, even ones that did not develop tumors, were included in the analysis.

Major comment 2a: In response to the provided images of invasive melanoma in zebrafish, the added explanation in pages 8-9 is satisfactory.

Major comment 2b: The authors acknowledge in Figure 3 that it is likely that the tumor is not a benign lesion and removed language suggesting such, which is satisfactory. Given that the tumor is not likely to be an MPNST because of its location, and is not benign given the pathological staining, authors could suggest that it is likely to be a melanoma.

Major comment 3a: The added text is good, but should be expanded, for clarity, to describe another mechanism such as transcriptional regulation.

Major comment 3b: In their discussion, the authors have added an additional hypothesis for ICER instability and tumor suppression which is acceptable.

Minor comment 1: The authors have included an additional western blot analysis that compares melanocytes to melanoma tumors; however, their methodology sorted mitfa+ cells as melanocytes. mitfa is expressed in zebrafish xanthophores, so these cells are also in the sample. To remedy this, the sample should be labeled as mitfa-positive rather than melanocytes.

Minor comment 2: The authors' revision is satisfactory.

Reviewer 2

Comments for the author

The authors have addressed my previous concerns and have improved the clarity and quality of the manuscript. I have no further comments.

Reviewer's Responses to Questions

Experimental quality

Does each figure have the proper controls?

If 'No', please indicate reasons in Comments for Author box below.

Reviewer #1:

- Yes

Reviewer #2:

- Yes
-

Were the data analyzed using appropriate statistical tests?

If 'No', please indicate reasons in Comments for Author box below.

Reviewer #1:

- Yes

Reviewer #2:

- Yes
-

Reproducibility

Were experiments performed using adequate number of biological replicates?

If 'No', please indicate reasons in Comments for Author box below.

Reviewer #1:

- Yes

Reviewer #2:

- Yes
-

Does the methods section provide sufficient detail to permit reproducibility?

If 'No', please indicate reasons in Comments for Author box below.

Reviewer #1:

- Yes

Reviewer #2:

- Yes
-

Completeness

Are the manuscript's conclusions supported by the data?

If 'No', please indicate reasons in Comments for Author box below.

Reviewer #1:

- Yes

Reviewer #2:

- Yes

Scholarship

Do the authors cite and discuss the merits of data that would argue for and against their conclusion?

If 'No', please indicate reasons in Comments for Author box below.

Reviewer #1:

- Yes

Reviewer #2:

- Yes

Does the manuscript title & abstract accurately reflect the contents of the manuscript, without hyperbole?

If 'No', please indicate reasons in Comments for Author box below.

Reviewer #1:

- Yes

Reviewer #2:

- Yes

Second revision

Author response to reviewers' comments

Reviewer 1: Comments pertaining to original review:

Major comment 1a: The authors do not directly address the possibility that S35-41A-ICER is not stably expressed; they should do so in the revised manuscript. Additionally, there is a discrepancy describing statistical significance in the results and discussion sections. In page 8 (line165) results the authors say that the difference between S35-41A ICER and EGFP lifespan was not statistically significant. However, in their discussion (page 8, line 333), authors claim that lifespan differences between all groups were statistically significant. The authors need to make this consistent. Add sentence that s35-41a icer is not stably expressed, as a possibility.

We now include the caveat that we have not been able to observe S35-41A-ICER protein, so its stable expression cannot be confirmed at this time.

Regarding the statistics, we thank the reviewer for pointing out the discrepancy. We have revised the text for clarity and consistency. Specifically, we now state that the overall survival difference across all three groups (EGFP, wt-ICER, and S35-41A ICER) was statistically significant, as determined by a log-rank (Mantel-Cox) test. However, in post-hoc pairwise comparisons using t-tests, we observed significant differences between EGFP and wt-ICER, as well as between S35-41A ICER and wt-ICER, but not between EGFP and S35-41A ICER. This clarification is now reflected in both the Results and Discussion sections, and we have specified the type of test used in each context.

Major comment 1b: The revised text pertaining to this comment needs clarification. It currently reads as though only animals with lesions were considered in the survival analysis. It should be clarified that all animals with rescued melanocytes, even ones that did not develop tumors, were included in the analysis.

The methodology in the text was what was performed, by first evaluating zebrafish that express pigment (indicating successful transgene integration via *mitfa* rescue). All zebrafish that had pigmentation rescue were followed, however only those that developed lesions were included in the survival study.

Major comment 2a: In response to the provided images of invasive melanoma in zebrafish, the added explanation in pages 8-9 is satisfactory.

No changes made.

Major comment 2b: The authors acknowledge in Figure 3 that it is likely that the tumor is not a benign lesion and removed language suggesting such, which is satisfactory. Given that the tumor is not likely to be an MPNST because of its location, and is not benign given the pathological staining, authors could suggest that it is likely to be a melanoma.

Language added in Figure 3 caption to suggest it is believed to be melanoma, while acknowledging the tumor type was not able to be confirmed.

Major comment 3a: The added text is good, but should be expanded, for clarity, to describe another mechanism such as transcriptional regulation

We thank the reviewer for pointing this out. The figure caption in Figure. 1 was updated to include the second western blot of FACS sorted *mitfa*⁺ cells, and acknowledges that mRNA transcript levels were not assessed, so transcriptional downregulation cannot be excluded. Our hypothesis of post-translational modification is supported by previous work that showed no reduction in mRNA levels of ICER in a RAS mutant model of melanoma compared with melanocytes, but reduced ICER protein expression in melanoma, with protein expression profile similar to our observations in this study.

Major comment 3b: In their discussion, the authors have added an additional hypothesis for ICER instability and tumor suppression which is acceptable.

No changes made.

Minor comment 1: The authors have included an additional western blot analysis that compares melanocytes to melanoma tumors; however, their methodology sorted *mitfa*⁺ cells as melanocytes. *mitfa* is expressed in zebrafish xanthophores, so these cells are also in the sample. To remedy this, the sample should be labeled as *mitfa*-positive rather than melanocytes.

We thank the reviewer for pointing this out. Figure 1B was updated to reflect that cells are *mitfa*⁺ rather than strictly melanocytes, as it is correct that xanthophore *mitfa* expression cannot be ruled out.

Minor comment 2: The authors' revision is satisfactory.

No changes made.

Reviewer 2: The authors have addressed my previous concerns and have improved the clarity and quality of the manuscript. I have no further comments.

No changes made.

Third decision letter

MS ID#: bio.061904R2

MS TITLE: Phosphorylation Deficient Inducible cAMP Early Repressor (ICER) Modulates Tumorigenesis and Survival in a Transgenic Zebrafish (*Danio rerio*) Model of Melanoma

AUTHORS: Justin Wheelan; Melissa Spigelman; Angelo Cirinelli; James Reilly; Carlos A. Molina

I am happy to tell you that your manuscript has been accepted for publication in Biology Open, pending our standard publication integrity checks. It was accepted on 14 July 2025.

Reviewer 1

Comments for the author

The authors have sufficiently addressed our comments.